# Fermentation of a Strong Dark Ale Hybrid Beer Enriched with Carob (*Ceratonia siliqua* L.) Syrup with Enhanced Polyphenol Profile

Katerina Pyrovolou [1], Panagiotis Tataridis [2], Panagiota-Kyriaki Revelou [3], Irini F. Strati [1], Spyros J. Konteles [1], Petros A. Tarantilis [3], Dimitra Houhoula [1] and Anthimia Batrinou [1,*]

1 Department of Food Science and Technology, School of Food Sciences, University of West Attica, 12243 Athens, Greece; apyrovolou@uniwa.gr (K.P.); estrati@uniwa.gr (I.F.S.); skonteles@uniwa.gr (S.J.K.); dhouhoula@uniwa.gr (D.H.)

2 Department of Wine, Vine and Beverage Sciences, University of West Attica, 12243 Athens, Greece; ptataridis@uniwa.gr

3 Laboratory of Chemistry, Department of Food Science and Human Nutrition, Agricultural University of Athens EU-CONEXUS European University, 11855 Athens, Greece; p.revelou@uniwa.gr (P.-K.R.); ptara@aua.gr (P.A.T.)

* Correspondence: batrinou@uniwa.gr

**Abstract:** There is an increasing trend to develop beers supplemented with local plant ingredients in order to increase their bioactivity. Carob (*Ceratonia siliqua* L.) is a xerophytic endemic tree typically found in Mediterranean ecosystems. The aim of this study was to develop a strong dark ale hybrid beer enriched with carob syrup prepared by using carob fruits from the University Campus (Athens, Greece). Three batches of beer were fermented, a dark ale (6% alcohol by volume or ABV) without carob and two strong dark ale beers (8% and 10% ABV) with carob syrup. After the second fermentation (bottle conditioning, 60 days), both carob beers had significantly increased bioactivity. The total phenolic content (176.4 mg GAE/100 mL), the antiradical activity (206.6 mg Trolox Equivalent (TE)/100 mL), and the antioxidant activity (838.2 mg $Fe^{2+}$/100 mL) of the carob strong dark ale 10% ABV beer was increased by more than three times, six times, and eight times, respectively, compared to the standard dark ale (6% ABV) without carob. Moreover, LC-QToF-MS analysis ascertained the enhancement of the phenolic profile of carob beers by ten phenolic compounds compared to the control dark ale beer without carob, indicating their significant antioxidant activity.

**Keywords:** strong dark ale; phenolic compounds; carob syrup; *Saccharomyces cerevisiae*; antioxidant activity; LC-QToF-MS analysis

## 1. Introduction

Beer is one of the most ancient and most widely consumed alcoholic beverages globally, contributing significantly to the economies of the producing countries [1,2]. In 2022, world beer production increased by 25 m hectoliters to 1.89 bn hectoliters, representing a growth of 1.3% [3]. Moreover, the beer industry experiences constant evolution and innovation driven by changing consumer preferences and demands. Craft breweries, which focus on small-scale specialty beer production, have surged in popularity in many markets [4]. These breweries often emphasize unique flavors, local ingredients, and traditional brewing techniques, appealing to consumers seeking distinct experiences [5,6]. Also, recent studies have focused on the development of beers that provide sensory characteristics combined with health benefits by adding plant ingredients such as aromatic herbs, spices, and fruits, which have increased bioactive compounds [7–9]. Supplementation of food products and beverages with antioxidants usually shows beneficial biological effects, preventing oxidative stress and reducing oxidative damage to cells. Polyphenols are considered strong antioxidants that can neutralize free radicals by donating an electron or hydrogen atom, and

they have been well-documented to counteract the degree of oxidative stress on cells and, as a consequence, to reduce the incidence of diseases associated with oxidative damage [10]. Fruits, vegetables, whole grains, and other types of foods and beverages, such as tea, chocolate, and wine, are rich sources of polyphenols, which are known for their health benefits [11–15]. Thus, it is believed that the administration of exogenous antioxidants through diet is beneficial [16]. Carob (*Ceratonia siliqua* L.) is an evergreen tree belonging to the family *Fabaceae* (or *Leguminosae*) and is a xerophytic endemic species characteristic of the Mediterranean biodiversity [17,18]. Several studies have shown the biological activity of carob components, including antioxidant, cytotoxic, and antidepressant activity [19,20]. Carob fruit is a unique product rich in dietary fiber content with a high concentration (1.7%) of polyphenols [21]. The main classes of phenolic compounds found in carobs are phenolic acids, gallotannins, and flavonoids, and their concentration in carob fruits is highly dependent on genetic, environmental, and extraction methods and ranges between 45 and 5376 mg of gallic acid equivalents per 100 g [22–24]. Therefore, producing new beer with carob syrup, which is a traditional carob product derived from naturally sweet carob fruit, could have several significant antioxidant and nutritional benefits. Also, from an ecological standpoint, the use of carob ingredients in commercial food and beverages may be linked to more sustainable and environmentally friendly practices since carob trees are resilient and drought-resistant, requiring minimal water and agricultural inputs [25]. Therefore, by incorporating carob syrup into beer production, breweries can promote more sustainable practices and contribute to a reduced ecological footprint. Another new trend in global beer production is the production of higher alcohol-by-volume (ABV) beers, which are brews that contain a more significant percentage of alcohol compared to standard beers [6]. While standard beers typically have an ABV from 4% to 6%, high-ABV beers can exceed 10% and sometimes even go beyond 15% [6]. The aim of this study is to develop a strong dark ale hybrid beer enriched with carob syrup, with higher alcohol-by-volume and increased polyphenol content.

## 2. Materials and Methods

### 2.1. Production and Treatment of Beer Samples Prior to Analysis

2.1.1. Preparation of Carob Syrup

Carob pods from the courtyard of the University of Western Attica campus (Athens, Greece) were utilized, along with carob pods sourced from Crete. Carob pods were washed and chopped into smaller pieces, approximately 2–3 cm in length, and the seeds were removed. The chopped carob pods were transferred to a pot with bottled water at a ratio of 1:2.4, and boiling was conducted at a temperature of 100 °C for 10 min. The resulting extract was cooled, placed in a membrane, and left at room temperature (~25 °C) for 2 to 3 days. Then, the extract was filtered as many times as necessary to remove all carob residues, leaving only the final extract in the pot. For optimal clarity of the final product, the extract underwent filtration using filter paper, and the process was repeated as needed, depending on the desired level of transparency. Finally, the final extract underwent re-boiling at 100 °C until sufficient condensation occurred, and the extract acquired the appropriate viscosity characteristic of the specific food category (Figure 1).

2.1.2. Preparation of Dark and Strong Dark Ale Carob Beers

For the production of 20 L of wort, the malts and other ingredients used and the mashing regime are presented in Table 1.

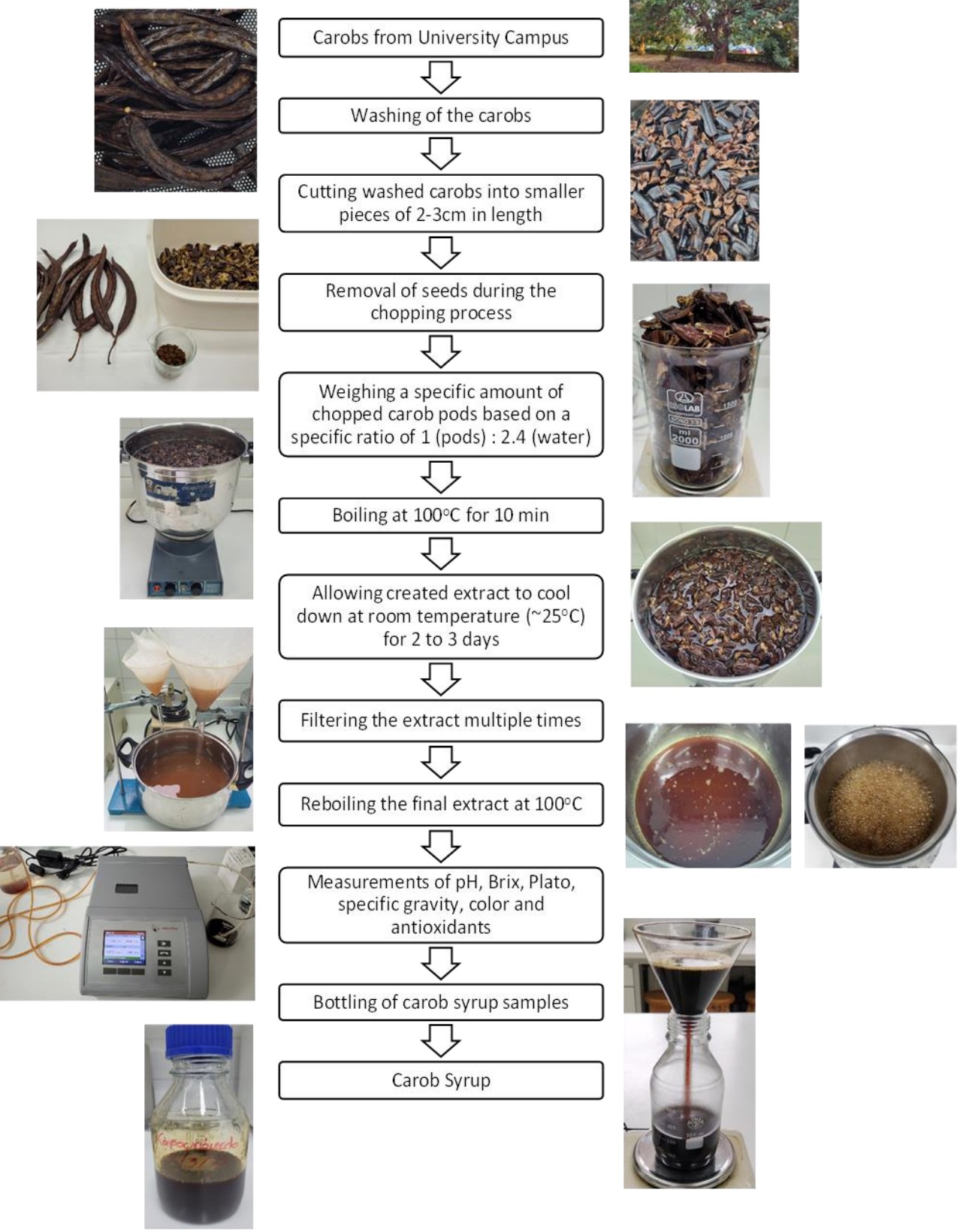

**Figure 1.** Flowchart of the preparation of carob syrup.

**Table 1.** (**A**) Ingredients of wort. (**B**) Mashing regime.

| (A) Ingredients | Type | # | %/IBU |
|---|---|---|---|
| Attica municipal water | Water | 1 | - |
| Pale Malt (2 Row) Vergina (5.5 EBC) | Grain | 2 | 75.0% |
| Vienna Malt (Weyermann) (9.0 EBC) | Grain | 3 | 6.7% |
| Caramunich II (Weyermann) (110.0 EBC) | Grain | 4 | 5.0% |
| Chocolate Malt (Simpsons) (1000.0 EBC) | Grain | 5 | 4.7% |
| Special W (Weyermann) (300.0 EBC) | Grain | 6 | 3.3% |
| Carafa Special I (Weyermann) (1000.0 EBC) | Grain | 7 | 2.8% |
| Wheat Malt, Pale (Weyermann) (3.9 EBC) | Grain | 8 | 2.4% |
| Northern Brewer [10.00%]—Boil 60.0 min | Hop | 9 | 48.4 IBUs |
| Irish Moss (Boil 10.0 min) | Fining | 10 | - |
| Cascade [7.80%]—Steep/Whirlpool 5.0 min | Hop | 11 | 3.4 IBUs |
| New World Strong Ale (Mangrove Jack's #M42) | Yeast | 12 | - |

| (B) Mash Steps | | | |
|---|---|---|---|
| **Name** | **Description** | **Step Temperature** | **Step Time** |
| Mash In | Add water at 69 °C | 65.0 °C | 60 min |
| Mash Step | heat to 72.0 °C over 8 min | 72.0 °C | 10 min |
| Mash Out | Heat to 75.6 °C | 75.6 °C | 10 min |

Sparge: Fly sparge with water at 75.6 °C.

Plato is defined as the concentration of dissolved solids in the extract per 100 g of solution, and the unit of measurement is Plato degrees (°P). A wort with characteristics of dark beer was prepared, and the initial Plato was adjusted to 13.8 °P. Subsequently, the wort was divided into 3 batches, each with a different final alcohol by volume (ABV). Initially, the 1st batch had no changes compared to the initial wort (control), and no quantity of carob syrup was added to it. Therefore, this particular beer had an expected final alcohol-by-volume of 6% ABV (coded as B) and was used as the "control" beer. In the 2nd and 3rd batches of wort, carob syrup was added in different proportions (100 g/L and 222 g/L in order to achieve degrees 18.8 °P and 23.8 °P respectively) to upgrade them to strong dark beers with higher final alcohol percentages. In the 2nd batch, the expected ABV was 8% (sample coded as CB1: strong dark ale carob beer 8% ABV), while in the 3rd batch, it was 10% (sample coded as CB2: strong dark ale carob beer 10% ABV). Carob syrup C (from ripe carobs collected from Attica, Greece) with initial Brix 71.35, prepared in the laboratory, was added to the final wort. For the dark ale beer with 6% ABV, 0.5 g/L of yeast was added. For the strong dark ale carob beer with 8% ABV, 0.7 g/L of yeast was added, and for the strong dark ale carob beer with 10% ABV, 0.86 g/L of yeast was added. No extra sugar was added in the first fermentation, and 6.5 g/L sugar was added in the second fermentation (bottle conditioning). During the fermentation process, analyses were conducted to scrutinize the samples for apparent contaminations. Organoleptic analyses, along with pH measurements, were performed, and the results revealed no indications of autolysis odors or alterations in the pH values of the samples. The procedure followed is shown in Figure 2.

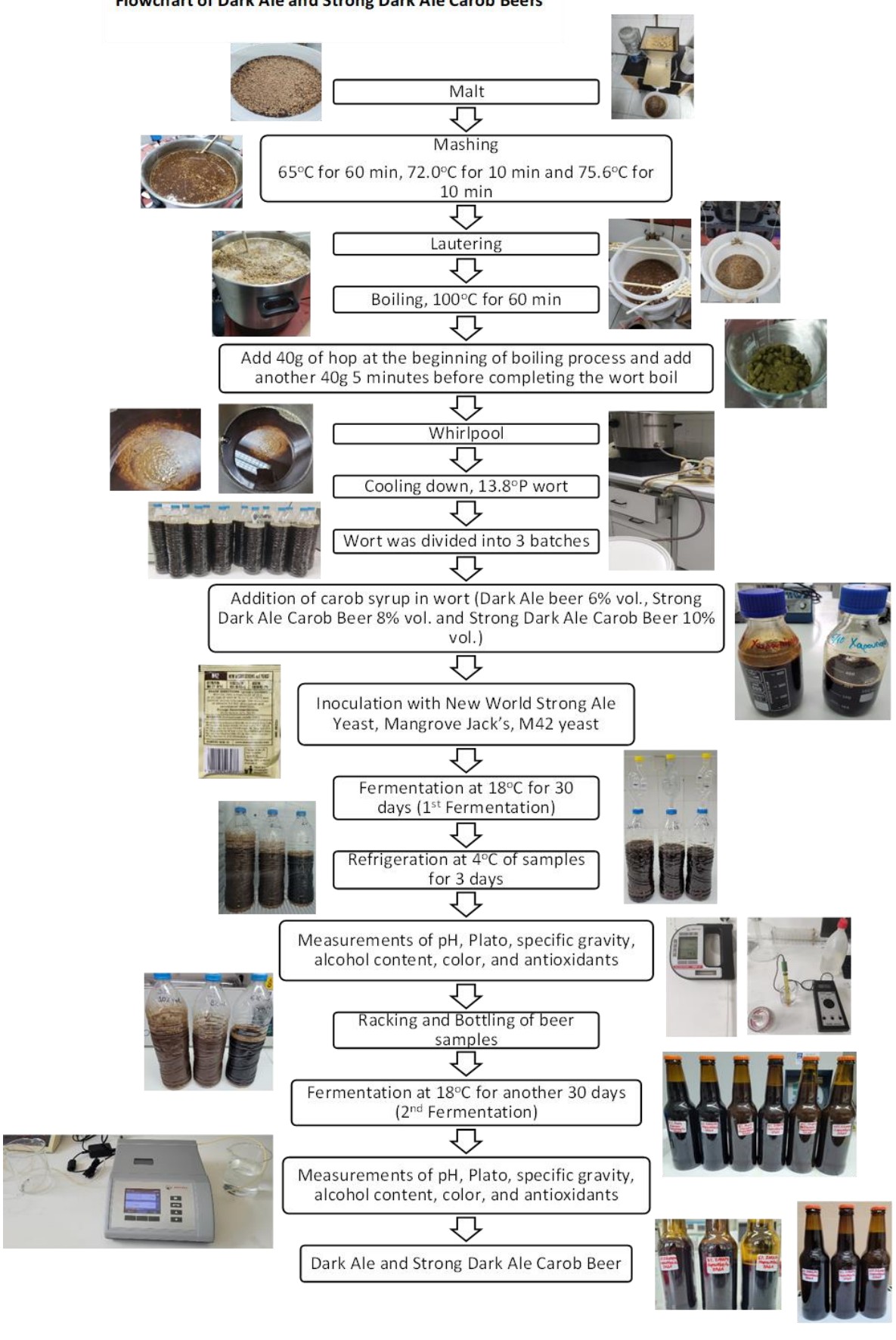

**Figure 2.** Flowchart of the preparation of dark ale beer and strong dark ale carob beers.

2.1.3. Sample Preparation for Beer Analysis

Prior to the analyses, beer samples undergo processing to ensure precision in measurements and results. Specifically, the method of protein precipitation was chosen for sample clarification [26]. The procedure involves transferring 8 mL of the sample and mixing it with 1 mL of $ZnSO_4$ (5%) and 1 mL of $Ba(OH)_2$ (0.3 N). The mixture was stirred and allowed to settle for 5 min. Afterward, re-stirring was performed, and the sample was left undisturbed for another 5 min to achieve the precipitation of aggregates. Subsequently, sample centrifugation was conducted at 3500 rpm for 10 min (MRC Laboratory Instruments Centrifuge), and the supernatant was collected and transferred to a new falcon tube. Due to the high turbidity of the samples and the presence of particles, post-centrifugation filtration was carried out using Whatman membrane filters nylon with a pore size of 0.45 μm and a diameter of 47 mm (Life Sciences).

*2.2. Analysis of Beer and Carob Syrup Samples*
2.2.1. Physicochemical Characteristics (pH, Brix, Plato, Specific Gravity, Alcohol Content, Color)

The determination of soluble solids for all carob syrup samples was performed using the XS instruments Digital Refractometer LDR-500-DRB 95 at room temperature. In detail, a few drops of the sample were uniformly placed on the prism of the refractometer, avoiding bubble formation, which could lead to deviations in value calculations. Brix units were used as the measurement unit of the refractometer, corresponding to the mass fraction of sucrose, where 1 Brix equaled 1 g of sucrose per 100 g of solution. Finally, pH was measured (Hanna instruments, HI 8010 pH meter). All measurements were conducted in triplicate to calculate the mean and standard deviation of the values. For beer samples, the measurements of pH (ASBC Method beer-9), specific gravity (SG), as well as Plato degrees (ASBC Method beer-2B and 3), were conducted throughout the fermentation. Specifically, samples were measured at the beginning of fermentation (t = 0 days), before bottling (t = 30 days, 1st fermentation), and after the completion of maturation (t = 60 days, 2nd fermentation). A crucial step before Plato and pH measurements is the deaeration, centrifugation, and filtration of samples. In detail, 80 mL of the sample is placed in falcon tubes and centrifuged at 3500 rpm for 10 min. Subsequently, samples are filtered through 0.45 μm filters to achieve the desired clarity, which is a necessary step before measurement. For Plato and specific gravity measurements, the handheld digital densitometer Anton Paar-DMA 35 was used, which takes samples through its integrated pump. The pH measurements were performed using the pH meter HANNA Instruments-HI 8010. All measurements were conducted in triplicate when samples were at ambient temperature (~25 °C). The measurements of alcohol content (ASBC Method beer-4), density (g/cm³), original extract % *w/w*, and real extract % *w/w* were carried out using the Anton Paar-Alex 500 instrument (alcohol and extract meter). The combination of absorption measurement via NIR spectroscopy and density measurement based on the oscillating U-tube principle constitutes the mechanism through which analysis is achieved by the instrument. The procedure preceding the sample analysis in the spectrophotometer is the same as described earlier. Therefore, sample clarification, achieved through centrifugation at 3500 rpm for 10 min and subsequent filtration through a 0.45 μm filter, is a necessary step before each analysis in the spectrophotometer. The results of the apparent final extract (°P) were used for the final calculation of the degree of fermentation and, consequently, the ethanol yield (% *v/v*). The measurement of color in beer samples (ASBC Method beer-10) was achieved through absorbance measurement at 430 nm and 700 nm using a single-beam spectrophotometer (Shimadzu–UVmini 1240 Model Spectrophotometer) in 1 cm cuvettes. Color measurement was performed at the beginning of the fermentation process, prior to bottling the samples, as well as on the final beers after maturation. The units of color measurement are expressed in EBC (European Brewery Convention). The sample is considered clear when the absorbance value at 700 nm ($A_{700nm}$) is less than or equal to the absorbance value at 430 nm ($A_{430nm}$) multiplied by 0.039. The measurement of the color value in EBC units is calculated by the following

formula: Beer Color EBC: $25 \times A_{430nm} \times D$, where D = dilution factor (in case dilution was performed on the initial sample).

### 2.2.2. Spectrophotometric Assays
### Determination of Total Phenolic Content (TPC)

The Total Phenolic Content (TPC) of the beer and carob syrup samples was determined using a modified version of the Folin–Ciocâlteu assay [27]. The absorbance was measured at 750 nm with a visible spectrophotometer (Spectro 23, Digital Spectrophotometer, Labomed Inc., Culver City, CA 90034, USA). The results were expressed as mg Gallic acid equivalents (GAE) per 100 mL of beer, using a standard curve with a range of 25–2600 mg/L Gallic acid ($y = 0.0005x + 0.0783$, $R^2 = 0.9989$).

### Ferric Reducing/Antioxidant Power Assay (FRAP)

In the ferric-reducing antioxidant power (FRAP) assay, antioxidants are evaluated as reductants of Fe(III) to Fe(II). The ferric-reducing antioxidant power assay, based on the reduction in a ferric-2,4,6-tripyridyl-s-triazine complex to the ferrous form, was carried out according to the method described by Lantzouraki et al. (2016) [28]. The absorbance was measured at 595 nm with a visible spectrophotometer (Spectro 23, Digital Spectrophotometer, Labomed Inc., Culver City, CA 90034, USA). A standard curve ($y = 0.0003x + 0.0081$, $R^2 = 0.9969$) was prepared using various concentrations (50–1800 μM) of $FeSO_4.7H_2O$ stock solutions. The results were expressed as mg Fe (II) per 100 mL of beer.

### Scavenging Activity on 2,2′-Azino-bis-(3-ethylbenzothiazoline-6-sulfonic Acid) Radical (ABTS$^{\bullet+}$)

The antiradical activity of beer and carob syrup samples was determined according to the method described by Lantzouraki et al. (2014) [29]. Absorbance was measured at 734 nm with a visible spectrophotometer (Spectro 23, Digital Spectrophotometer, Labomed Inc., Culver City, CA 90034, USA). Trolox, a water-soluble form of vitamin E, was used as a standard compound, and the antiradical activity of each sample was expressed as mg Trolox Equivalents (TE) per 100 mL of beer. A standard curve was prepared with a concentration range of 0.20–1.50 mM Trolox ($y = 0.2876x - 0.002$, $R^2 = 0.9995$).

### 2.3. Phenolic Profile by LC-QToF-MS Analysis
### 2.3.1. Reagents and Materials

Acetonitrile and acetic acid (Merk KGaA, Darmstadt, Germany) were of LC-MS grade. A Genie Water System (RephiLe Bioscience Ltd., Shanghai, China) was used to obtain ultrapure water. Phloroglucinol, gallic acid, protocatechuic acid, gentisic acid, (-)-catechin, 4-hydroxybenzoic acid, vanillic acid, epicatechin, syringic acid, p-coumaric acid, ferulic acid, myricitrin, absisic acid, trans-cinnamic acid, luteolin, and naringenin were acquired from Extrasynthese (Genay, France).

### 2.3.2. Mass Spectra Analysis

The mass spectra of samples were acquired using an Agilent 6530 Quadrupole Time of Flight (QToF) system equipped with an electrospray ionization (ESI) interface (Agilent Technologies, Santa Clara, CA, USA). The QToF system was coupled with a UHPLC system (Agilent 1290 Infinity) and an autosampler obtained from Agilent Technologies (Santa Clara, CA, USA). The mass spectrometer was operated at negative ion mode (-ESI). The following parameters were applied: capillary voltage 4000 V; nebulizer gas pressure 45 psig; drying gas flow rate 10 L/min; gas temperature 300 °C; fragmentor; and skimmer voltages 150 V and 65 V, respectively. For the MS/MS experiments, an auto MS/MS method was applied with the following parameters: collision energy slope, 5 V; offset, 2.5 V; acquisition rate of MS/MS, 1 spectra/s; preferred charge state, 2, 1, unknown. The QToF-MS system was calibrated before the analysis using a reference solution acquired from Agilent Technologies (Santa Clara, CA, USA). A constant infusion of a reference mass solution with

ions 112.9856 and 1033.9881 was applied during this analysis. The Agilent MassHunter Data Acquisition software (version B.06.00) was used for data acquisition. A Nucleoshell EC C18 (100 mm × 4.6 mm, 2.7 μm) column (Macherey-Nagel GmbH & Co., Düren, Germany) was used for the chromatographic study. The mobile phase consisted of (A) ultrapure water–acetic acid 0.1% and (B) acetonitrile–acetic acid 0.1%, with a flow rate of 1.0 mL min$^{-1}$. The gradient program was as follows: 0 min: 10% B; 8 min: 30% B; 12 min: 40% B; 16 min: 50% B; 18 min: 10% B; and 33 min: 10% B, with a total run time of 33 min. The injection volume was 10 μL, while the column temperature was set at 30 °C. Data processing was performed using the Agilent MassHunter Qualitative Analysis software (version B.07.00). Compound identification was based on the retention time relative to that of the standards, the accurate mass of ions, and their MS/MS spectra. The identification of compounds for which there were no standards available was based on the Riken Tandem Mass Spectral Data Base library database (http://spectra.psc.riken.jp/, accessed on 1 May 2023).

### 2.4. Statistical Analysis of Results

For all analyses, three measurements were conducted to calculate the average values and standard deviation. The data regarding total phenolic content, antiradical, and antioxidant activity were analyzed with One-Way ANOVA Post Hoc Tests, using the Duncan discrimination test with statistical significance ($p < 0.05$). The correlation among the results was performed by the Spearman rank order test. Statistical analysis was performed using the Statistica package (STATISTICA version 7.0, software Statsoft Inc., Tulsa, OK, USA, 2004).

## 3. Results and Discussion

### 3.1. Carob Syrup

The carob pulp has a high concentration of sugars (48–56%), mainly sucrose, glucose, and fructose [18], and is used in many Mediterranean countries to prepare traditional syrups. Four syrups were prepared in the lab, three from carobs collected from trees grown on the University campus (Egaleo, Athens, Greece) and one from carobs of Crete (Table 2). The carob syrups produced in the laboratory were viscous and chocolate-colored syrups. According to the literature references, carob syrup contains solid solutes ranging from 62 to 74 Brix [30], and the Brix degrees of the samples prepared in the laboratory ranged from 64.30 to 74.43 (Table 2). The specific differentiation in Brix degrees is likely due to the final boiling time, which is also responsible for the final concentration degree of the product. The pH values of the carob syrups analyzed in this study ranged from 4.91 to 5.91 and are in agreement with similar studies that have found the pH value of carob syrup ranged from 4.3 to 5.4 [31]. Organic acids, as well as minerals contained in syrups, mainly account for their acidic pH.

**Table 2.** Measurements of pH and Brix of carob syrups prepared in the laboratory and commercially available carob syrup.

| Sample | pH | Brix |
|---|---|---|
| Carob syrup A (Uniwa) | 5.08 | 67.80 |
| Carob syrup B (Uniwa) | 4.91 | 64.30 |
| Carob syrup C (Uniwa) | 5.08 | 71.35 |
| Carob syrup D (Crete) | 5.41 | 74.43 |
| Commercial Carob syrup | 5.31 | 72.61 |

### 3.2. Characteristics of Dark and Strong Dark Ale Beer Samples

The purpose of this study was to brew a dark ale beer, which was upgraded to a strong dark ale beer by adding carob syrup. The difference between the two beer categories is that a dark ale has lower final alcohol by volume (ABV), while when upgraded to a strong dark ale, the expected ABV significantly increases. According to the process of alcoholic fermentation, the yeasts added to the beers metabolize the sugars present in the initial

malts, resulting in the production of ethanol and carbon dioxide as the final fermentation products. Therefore, when carob syrup is used as an additive to the initial beer, it provides a higher concentration of sugars and an increase in Plato, which can be consumed by yeasts, producing higher alcohol percentages. Thus, through the availability of sugars, the initial dark ale is enriched and upgraded to a strong dark ale beer. Simultaneously, in addition to providing carbohydrates, carob syrup enriches the final products with other equally important components, such as polyphenols. Typically, dark malts rich in sugars and other ingredients are used in stouts and dark ale beers, giving the final product an intense flavor, characteristic aroma, and dark color [32]. At the beginning of the beer fermentation process (t = 0 days), the Plato values for the dark ale beer with 6% ABV were 13.8 °P, for the strong dark ale carob beer with 8% ABV, they were 18.8 °P, and finally, for the strong dark ale carob beer with 10% ABV, they were 23.8 °P (Table 3). The consumption of fermentable sugars resulted at the end of the first fermentation process (t = 30 days), at Plato 3.8 °P for the dark ale beer with 6% ABV, 4.8 °P for the strong dark ale carob beer with 8% ABV, and finally, 6.3 °P for the strong dark ale carob beer with 10% ABV. No significant changes were expected at the final Plato values at the end of the second fermentation since only a small amount of dextrose (6.5 g/1 L of beer) was added, and it was mostly a product maturation step (bottle conditioning) in order to generate carbon dioxide in the final products. Therefore, for the dark ale beer with 6% ABV, the apparent fermentation degree was 72.46%, for the strong dark ale carob beer with 8% ABV, it was 74.46%, and finally, for the strong dark ale carob beer with 10% ABV, it was 73.52%, at the end of the first fermentation. Based on the fermentable sugars present in both the wort and carob syrups, these percentages are deemed acceptable [33]. The actual final % ABV of beers was very close to the expected: 5.57% for the control beer (expected 6%); 7.97% for the carob beer CB1 (expected 8%); and 9.97% for carob beer CB2 (expected 10%), values, which align with the limits specified in the literature for dark ale and strong dark ale beers (Brewers Association Beer Style Guidelines, n.d.). In all three samples, a decrease in pH was observed upon completion of the first fermentation (Table 3). The main cause was the production of weak organic acids (mainly lactic acid) during the fermentation of the samples, which affected the final pH value. According to the literature, the pH values for wort range from 5.3 to 5.6, while the pH limits for the final beer are from 4.3 to 4.6 [34]. Regarding sugars, in the dark ale beer with 6% ABV alcohol, all sugars come from the malt extracted during the mashing process. However, in the case of strong dark ale carob beers with 8% ABV and 10% ABV alcohol, sugars come from both malt and carob syrup added to the wort. The added yeast ferments these sugars, resulting in the production of the two main metabolites of alcoholic fermentation: ethanol; and carbon dioxide ($CO_2$). Consequently, as the alcohol concentration (% $v/v$) increases in beers, the sugar content (Plato) of the final product decreases, demonstrating an inversely proportional relationship.

In all three samples, a decrease in pH was observed upon completion of the first fermentation due to the production of weak organic acids during the fermentation. The final pH ranged from 4.43 to 5.07, while the pH limits for the commercial beers ranged from 4.3 to 4.6. (Silva et al., 2022). Moreover, from these results, it was concluded that carob beers were darker than traditional beers in the stout and imperial stout categories, according to the literature, and the reason was the use of dark malts in brewing and the addition of carob syrup to the final wort.

Another important factor is the measurement of the color of beer samples. Two main methods are used for color determination: the Standard Reference Method (SRM), established by the American Society of Brewing Chemists (ASBC); and the Color Units EBC (European Brewery Convention), established by the European Brewery Convention. In both methods, the color of beer samples is analyzed using a spectrophotometer at a wavelength of 430 nm. The primary difference between the two methods is the thickness of the cell used during analysis. In the SRM method, the sample is placed in a cell with a thickness of ½ inch, while in the EBC method, a cell with a thickness of 1 cm is used. The equation connecting the two methods and converting EBC units to SRM and vice versa is

as follows: SRM = EBC × 0.508. Figure 3 illustrates beer shades according to their category and the corresponding values for both methods [35]. To express the results of the EBC method, the following equation is applied: EBC = $A_{430}$ × 25 × Dilution factor. The dilution factor is proportional to the sample treatment and depends on whether reactants have been added to it for the precipitation of proteins. Due to the dark color of carob beers, a 1:6 dilution of samples was required, regardless of the additional dilution applied due to clarity. From these results, it was concluded that carob beers were darker than traditional beers in the dark (stout) and strong dark (imperial stout) categories, and the reason was the use of dark malts in brewing and the addition of the chocolate-colored carob syrup to the final wort.

**Table 3.** Physicochemical parameters of beer samples at the beginning of fermentation (t = 0 days), after the 1st (t = 30 days), and 2nd fermentations (t = 60 days).

| | t = 0 Days | | | 1st Fermentation (t = 30 Days) | | | 2nd Fermentation (t = 60 Days) | | |
|---|---|---|---|---|---|---|---|---|---|
| | **B: Dark Ale Beer (6% ABV)** | **CB1: Strong Dark Ale Carob Beer (8% ABV)** | **CB2: Strong Dark Ale Carob Beer (10% ABV)** | **B: Dark Ale Beer (6% ABV)** | **CB1: Strong Dark Ale Carob Beer (8% ABV)** | **CB2: Strong Dark Ale Carob Beer (10% ABV)** | **B: Dark Ale Beer (6% ABV)** | **CB1: Strong Dark Ale Carob Beer (8% ABV)** | **CB2: Strong Dark Ale Carob Beer (10% ABV)** |
| pH | 5.24 | 5.27 | 5.26 | 4.56 | 4.93 | 5.16 | 4.43 | 4.79 | 5.07 |
| °plato | 13.8 | 18.8 | 23.8 | 3.8 | 4.8 | 6.3 | 3.1 | 4.3 | 6.5 |
| Alcohol (% *v/v*) | 0% | 0% | 0% | 5.42 | 7.97 | 10.31 | 5.57 | 7.97 | 9.97 |
| Degree of fermentation % | n/a | n/a | n/a | 72.46 | 74.46 | 73.52 | | | |
| EBC | 127.69 | 133.87 | 92.75 | 119.35 | 129.94 | 120.13 | 88.75 | 95.25 | 101.69 |
| SRM | 64.86 | 68.01 | 47.12 | 60.63 | 66.01 | 61.02 | 45.08 | 48.39 | 51.66 |

**Figure 3.** Values of the SRM and EBC methods and their color representation (Standard Reference Method, 2021).

The turbidity of beer samples is another factor examined in addition to color. The determination of turbidity is calculated in the spectrophotometer based on the absorption of beer samples at wavelengths 430 nm and 700 nm, using the ASBC Turbidity method. A beer sample is characterized as clear (free of turbidity) when the following condition is met: $A_{700nm} \leq 0.039 \times A_{430nm}$. In the case where the absorption at 700 nm is higher, the sample is assessed as turbid. The beer samples prepared in the laboratory (dark ale 6% ABV, strong dark ale carob beer 8% ABV, and strong dark ale carob beer 10% ABV) were evaluated as turbid, according to this specific method.

### 3.3. Total Phenolic Content (TPC), Antioxidant Activity, and Antiradical Activity

All carob syrup samples prepared exhibited a high content of total polyphenols. This is primarily due to the high concentration of polyphenols and constituents present in

the carob used for the preparation of the carob syrups [22,36,37]. The carob syrup used to prepare the carob beers had a total phenolic content of $911.4 \pm 19$ mg GAE/100 mL, antioxidant capacity expressed as $6464 \pm 69.6$ mg $Fe^{2+}$/100 mL, and antiradical activity $2514 \pm 45.3$ TE/100 mL. Similar studies have determined a mean value of $729 \pm 487$ g GAE/100 g among 14 carob syrups [31] and 9.51–14.80 mg GAE/g among nine carob syrups from Cyprus and Greece [36]. The high concentration of polyphenols makes the carob syrups ideal as additives and capable of enriching foods and beverages with antioxidants.

Beer presents significant antioxidant activity mainly attributed to the phenolic compounds originating from barley and hops [38–40]. The dark ale beer 6% ABV (B-60d) produced in this study had a total phenolic content of 48.9 mg GAE/100 mL at the end of the second fermentation (Figure 4). Similar studies have shown that dark beers had a minimum TPC content of 300 mg GAE/L and that dark beers had higher TPC than pale beers due to the greater presence of malted barley and the generation of polyphenols during the malting process [41]. However, it was shown that the addition of carob syrup significantly increased the bioactivity of carob beers. The strong dark ale carob beers had significantly higher total phenolic content, antioxidant activity, and antiradical activity compared to the dark ale beer without carob (Figure 4). More specifically, the carob strong dark ale with 8% ABV (CB1), after the second fermentation (60 days), had total phenolic content (99.9 mg GAE/100 mL), antiradical activity (129.0 mg TE/100 mL), and antioxidant activity (387.0 mg $Fe^{2+}$/100 mL), which was higher by approximately two times, four times, and three times, respectively, compared to the respective values of the standard dark ale (6% ABV) without the addition of carob syrup (48.9 mg GAE/ 100 mL, 31.7 mg TE/100 mL and 105.4 mg $Fe^{2+}$/ 100 mL). The carob strong dark ale with 10% ABV (CB2) after 60 days of fermentation had total phenolic content (176.4 mg GAE/100 mL), antiradical activity (206.6 mg TE/100 mL), and antioxidant activity (838.2 mg $Fe^{2+}$/100 mL), which was higher by more than three times, six times, and eight times, respectively, compared to the respective values for the standard dark ale (6% ABV) without the addition of carob syrup.

Many studies have shown that the addition of plant ingredients rich in bioactive substances during craft beer brewing might increase the total phenolic content and antioxidant activity of the final product [7–9,42,43]. The contents of polyphenolic compounds and the ABTS and DPPH radical scavenging abilities were increased by more than 2.0-fold, 2.0-fold, and about 6.0-fold, respectively, in beers brewed with the addition of dotted hawthorn (*Crataegus punctata*) juice [8]. In another study, the total polyphenol contents and the ABTS and DPPH radical scavenging activities in beer were increased by 42.8%, 44.3%, and 42.4%, respectively, with the addition of mango juice [42]. Moreover, high levels of total phenolic compounds and antioxidant activity were achieved by the replacement of hops with rubim (*Leonurus sibiricus* L.) and mastruz (*Chenopodium ambrosioides* L.) [44]. Most similar studies have analyzed the polyphenols of carob syrups and carob pulp extracted usually in various organic solvents rather than in water, whereas in this study the polyphenols were extracted during the traditional preparation of carob syrup with water. Also, although a thermal treatment was applied, the total phenolic content was found to be high in the final product, whereas other studies had observed degradation of the phenolic content of carob syrups [36]. Nevertheless, some variations were observed after the second fermentation, such as a decline in antioxidant and antiradical activity; however, the overall activities remained significantly higher than the control beer (without carob). Similar results were obtained by other researchers (Ditrych et al., 2015), who have studied the antiradical and reducing potential of commercial beers during storage and observed a decrease in antioxidant activity as a result of storage, occurring mainly after the initial 4-week storage period. A possible explanation for this decrease could be that the level of oxygen is not easily controlled; therefore, the beers should be consumed rather shortly, and, moreover, the produced beers are unpasteurized. Additionally, He et al. (2012) observed 10% lower antioxidant activity in fresh cloudy wheat beer over the first 18 days of storage at 20 °C.

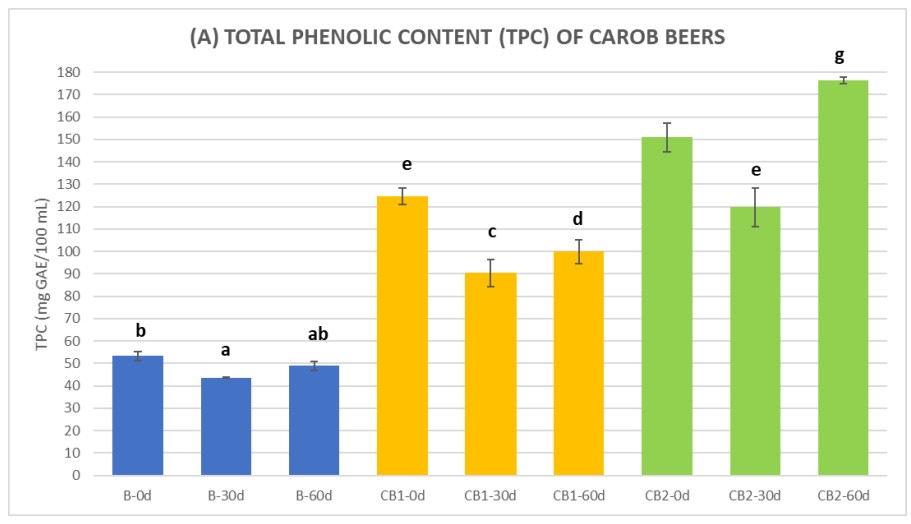

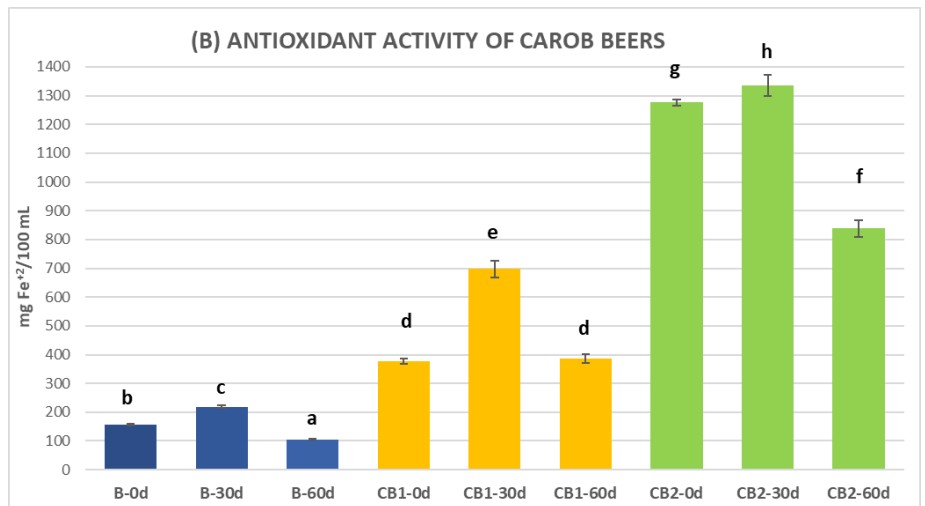

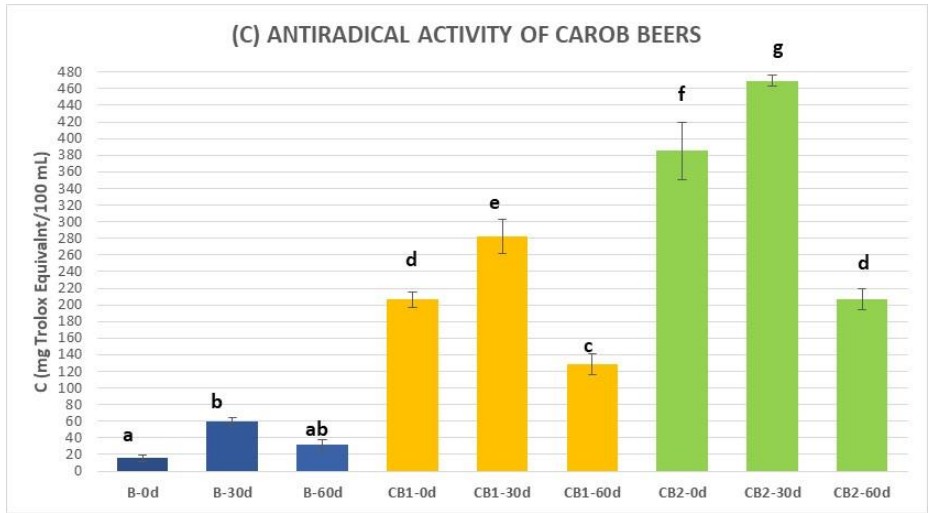

**Figure 4.** (**A**) Total phenolic content of samples (expressed as mg GAE/100 mL). (**B**) Antioxidant activity (expressed as mg Fe$^{+2}$/100 mL). (**C**) Antiradical activity (mg TE/100 mL). Standard dark ale beer with 6% vol without carob used as control (**B**). Strong dark ale carob beer with 8% ABV (CB1) and strong dark ale carob beer with 10% vol (CB2) at the beginning of fermentation (0 d), after the 1st fermentation (30 d) and 2nd fermentation (60 d). Bars bearing different letters are significantly statistically different ($p < 0.05$) (a < b < c < d < e < f < g < h).

Furthermore, a strong relationship was observed between radical scavenging activity measured by ABTS assay and ferric-reducing/antioxidant power assay (FRAP) results, with a correlation coefficient $R^2$ of 0.94 ($p < 0.05$). The correlation coefficient $R^2$ ($p < 0.05$) between the total polyphenol content (TPC), ferric-reducing/antioxidant power assay (FRAP), and radical scavenging activity (ABTS assay) was found to be 0.77 and 0.68, respectively, indicating that polyphenols contributed significantly to the reducing/antioxidant power of beers [45,46].

### 3.4. LC-QToF-MS Analysis

Analysis of phenolic compounds has identified polyphenols in the control sample (B = dark ale 6% ABV without carob), the carob beers (CB1, CB2), and carob syrup (C), as shown in Table 4. However, the carob beers were enriched with 10 more phenolic compounds, which were all ingredients of carob syrup, the initial raw material used to produce the carob beer. Among phenolic acids, LC-QToF-MS analysis revealed the presence of hydroxybenzoic acids, such as gallic acid, 4-hydroxybenzoic acid, gentisic acid, protocatechuic acid, syringic acid, vanillic acid, and derivatives of hydroxycinnamic acid, such as p-coumaric acid, ferulic acid, absisic acid, and *trans*-cinnamic acid. Phenolic acids are the aromatic secondary metabolites imparting color, flavor, astringency, and harshness, which contribute to the typical organoleptic characteristics of foods and are comprised of one-third of the constituents among phenolic compounds [47].

**Table 4.** Phenolic compounds identified in beers and carob syrup by LC-QToF-MS.

| Compound | Sample | [1] $t_R$ (min) | Molecular Formula | Theoretical *m/z* [M-H]$^-$ | Experimental *m/z* [M-H]$^-$ | Mass Error |
|---|---|---|---|---|---|---|
| Phloroglucinol | [2] CB1, [3] CB2, [4] C | 1.62 | $C_6H_6O_3$ | 125.0244 | 125.0243 | 0.94 |
| Gallic acid | CB1, CB2, C | 1.64 | $C_7H_6O_5$ | 169.0142 | 169.0137 | 3.24 |
| Protocatechuic acid | [5] B, CB1, CB2, C | 2.39 | $C_7H_6O_4$ | 153.0193 | 153.0198 | −3.05 |
| Gentisic acid | CB1, CB2, C | 3.26 | $C_7H_6O_4$ | 153.0193 | 153.0188 | 3.50 |
| (-)-catechin | B, CB1, CB2, C | 3.47 | $C_{15}H_{14}O_6$ | 289.0718 | 289.0707 | 3.67 |
| 4-hydroxybenzoic acid | B, CB1, CB2, C | 3.54 | $C_7H_6O_3$ | 137.0244 | 137.0240 | 3.05 |
| Vanillic acid | B, CB1, CB2, C | 4.08 | $C_8H_8O_4$ | 167.0350 | 167.0350 | 0.00 |
| Epicatechin | B, CB1, CB2, C | 4.32 | $C_{15}H_{14}O_6$ | 289.0718 | 289.0714 | 1.25 |
| Syringic acid | B, CB1, CB2, C | 4.39 | $C_9H_{10}O_5$ | 197.0455 | 197.0457 | −0.77 |
| p-coumaric acid | B, CB1, CB2, C | 5.54 | $C_9H_8O_3$ | 163.0400 | 163.0392 | 4.97 |
| Ferulic acid | B, CB1, CB2, C | 6.31 | $C_{10}H_{10}O_4$ | 193.0506 | 193.0510 | −1.92 |
| Myricitrin | CB1, CB2, C | 6.48 | $C_{21}H_{20}O_{12}$ | 463.0882 | 463.0863 | 4.10 |
| Taxifolin | CB1, CB2, C | 6.57 | $C_{15}H_{12}O_7$ | 303.0510 | 303.0515 | −1.65 |
| Quercetin-3-glucoside | B, CB1, CB2, C | 6.74 | $C_{21}H_{20}O_{12}$ | 463.0882 | 463.0867 | 3.24 |
| Absisic acid | CB1, CB2, C | 9.05 | $C_{15}H_{20}O_4$ | 263.1289 | 263.1294 | −1.81 |
| trans-cinnamic acid | CB1, CB2, C | 9.79 | $C_9H_8O_2$ | 147.0451 | 147.0449 | 1.36 |
| Quercetin | CB1, CB2, C | 10.25 | $C_{15}H_{10}O_7$ | 301.0354 | 301.0346 | 2.66 |
| Luteolin | CB1, CB2, C | 10.33 | $C_{15}H_{10}O_6$ | 285.0405 | 285.0397 | 2.67 |
| Naringenin | CB1, CB2, C | 11.44 | $C_{15}H_{12}O_5$ | 271.0612 | 271.0613 | −0.37 |

[1] $t_R$: retention time. [2] CB1: imperial stout beer with 8% ABV. [3] CB2: imperial stout beer with 10% ABV. [4] C: carob syrup. [5] B: stout beer with 6% ABV.

Gallic acid, which was found only in carob beers, is a strong antioxidant and free radical scavenger that can protect biological cells, tissues, and organs from damage caused by oxidative stress. There are diverse reports with regard to gallic acid for medicinal uses, such as antibacterial, anti-allergy, anti-inflammatory, and antioxidant stress [48,49]. Other recent studies have identified gallic acid to be the most abundant phenolic compound in both ripe and unripe carob pulp extracts, and flavonoids such as myricetin, quercetin, and naringenin were detected [50]. Moreover, gallic acid was found to be the major component of carob syrups, which was probably co-extracted with carbohydrates during the preparation of carob syrup, while other polyphenols were detected in minor amounts [30,51]. In addition, other phenolic acids identified in the carob beers were gentisic acid, which is

a phenolic acid associated with antioxidant but also anti-inflammatory, hepatoprotective, antigenotoxic, antimicrobial, and neuroprotective activities [52], as well as trans-cinnamic acid which is a phenylpropanoid with a broad spectrum of biological activities, including antioxidant and antibacterial activities [53]. The flavonoids which were detected in this study were myricitrin, quercetin, quecertin-3-glucoside, naringenin, taxifolin, catechin, epicatechin, and luteolin. Among these, five were detected only in the carob beer samples: (a) myricitrin, a naturally occurring polyphenol hydroxy flavonoid, which has been reported to possess anti-inflammatory properties [54]; (b) quercetin, which is known as an antioxidant, anti-inflammatory, cardioprotective, and anti-obesity compound, also thought to be beneficial against cardiovascular diseases, cancer, diabetes, neurological diseases, obesity, allergy asthma, and atopic diseases [55]; (c) naringenin, which is a flavonoid belonging to flavanones subclass with several biological activities such as antioxidant, antitumor, antibacterial, antiviral, anti-inflammatory, antiadipogenic, and cardioprotective effects [56]; (d) taxifolin (dihydroquercetin), which is a powerful antioxidant with antioxidant, anti-inflammatory, anti-microbial, and other pharmacological activities [57]; and (e) luteolin, which is an important natural polyphenol present in several plants that show antioxidant, anti-inflammatory, anticancer, cytoprotective, macrophage polarization, and neuroprotective effects [58,59]. Finally, another phenolic compound identified in the carob beers was phloroglucinol. Phloroglucinol derivatives are a major class of secondary metabolites that can be classified into monomeric, dimeric, trimeric, higher phloroglucinols, and phlorotannins [60]. Phloroglucinols are known for their broad-spectrum antiviral, antibacterial, antifungal, antihelminthic, and phytotoxic activities [61].

The phenolic profile in a similar study was composed of p-coumaric acid, epigallocatechin gallate, epigallocatechin, catechin, syringic acid, quercetin glycoside, caffeic acid, gallic acid, catechin gallate, myricetin 3-glycoside, and cinnamic acid in acetone extracts from carob syrups analyzed by using HPLC-DADeESI-MS technique [30]. In another study, in ripe pulp extract, ultrasound-assisted extraction of polyphenols from carob pulp revealed a phenolic profile, which was dominated by gallic acid (65%) but also catechin, naringenin, cinnamic acid, quercetin, catechol, ferulic acid, gentisic acid, and gallic acid were detected [37]. Myricetin, catechin, epicatechin, quercetin, rutin, and syringic acid were found in the acidic acetone and acetone–water extracts of carob fruits extracted with various extraction solvents [62]. In an Egyptian study, aqueous extracts of carob pods contained a high content of gallic acid followed by catechin, protocatechuic acid, cinnamic acid, and a low concentration of p-coumaric acid, rutin, gentisic acid, p-hydroxybenzoic acid, vanillic acid, and ferulic acid [63]. Aqueous decoction of carob kibbles was found to be rich in antioxidants such as gallic acid, catechin, epigallocatechin gallate, gallocatechin gallate, and epicatechin [64].

## 4. Conclusions

In this study, two strong dark ale beers (8% and 10% ABV) were produced with the addition of carob syrup prepared in the University Laboratory from the fruits of carob trees grown on the Campus of the University of West Attica (Athens, Greece). The carob beers produced had the characteristics of strong dark ale (alcoholic content, density, degrees of Plato, and color) and also had significantly higher bioactivity compared to the standard dark ale with 6% ABV without carob. The stronger dark ale carob beer (10%), which also had a double quantity of carob syrup than the dark ale carob beer (8%), had the highest values of phenolic content (176.4 mg GAE/ 100 mL), antiradical activity (206.6 mg TE/100 mL), and antioxidant activity (838.2 mg $Fe^{2+}$/100 mL), which were increased by more than three times, six times, and eight times, respectively, compared to the standard dark ale (6% ABV) without carob. Moreover, analysis of polyphenols by LC-QToF-MS has revealed 10 more phenolic compounds (phloroglucinol, gallic acid, gentisic acid, myricitrin, taxifolin, abscisic acid, *trans*-cinnamic acid, quercetin, luteolin, and naringenin) in the carob beers compared to the dark ale beer without carob, indicating that the produced carob beers had an enhanced polyphenol profile. By producing beer with carob syrup,

breweries could not only develop a biofunctional product but also support local agriculture in regions where carob trees grow abundantly and are unexploited. Carob syrup is also naturally gluten-free, making it an excellent ingredient for producing gluten-free beers [65]. In conclusion, producing new beer with carob syrup offers a range of advantages, including increased bioactivity and sustainability.

**Author Contributions:** Conceptualization, A.B. and P.T.; methodology, K.P., P.-K.R., I.F.S. and P.A.T.; formal analysis, K.P., P.-K.R. and I.F.S.; data curation, K.P., A.B., S.J.K. and D.H.; writing—original draft preparation, K.P. and A.B.; writing—review and editing, K.P., A.B., P.T. and S.J.K.; supervision, A.B. All authors have read and agreed to the published version of the manuscript.

**Funding:** This research received no external funding.

**Institutional Review Board Statement:** Not applicable.

**Informed Consent Statement:** Not applicable.

**Data Availability Statement:** Data are available upon request.

**Conflicts of Interest:** The authors declare no conflicts of interest.

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
