# Peer review of "Fermentation of a Strong Dark Ale Hybrid Beer Enriched with Carob (Ceratonia siliqua L.) Syrup with Enhanced Polyphenol Profile"

_applsci, doi:10.3390/app14031199_

Round 1
Reviewer 1 Report
Comments and Suggestions for Authors
The addition of different types of plants, fruits or extracts is one of the current trends in beer production. Although this type of beverage does not fall under the definition for beer (according to the German beer purity law), this type of product forms a new range of beverages that will probably soon be defined as a specific style of beer. In this sense, the use of the word "hybrid" in the title is correct.
The paper itself is interesting. To the authors, I have the following questions, notes and suggestions for improving the publication:
1. Is the resulting syrup microbiologically stable, since it is described in materials and methods that it is left at room temperature?
2. What is the fermentable sugar content of the syrup? This is important from the point of view of the selected doses to increase the fermentable extract of the wort.
3. Please describe in detail the mashing regime for obtaining the wort.
4. Please specify the fermentable and non-fermentable wort extract.
5. The conditions of inoculation with the selected yeast strain are not specified. Please motivate the selected fermentation mode. 30 days at 18 °C is too long a fermentation that can lead to microbiological problems of the resulting beer!? Followed by cold storage, addition of sugar and secondary fermentation again for 30 days? Such a regime at relatively high temperatures may imply autolysis processes that would disturb the sensory profile of the beer.
6. The publication abounds in theoretical descriptions, but lacks comments on observed trends. For example, the description of what a Plateau is, what are the methods of determining color, etc. are redundant and can be replaced by the corresponding trend comments in Table 2.
7. The following questions arise from table 2:
• What is the reason for such a low fermentation degree in the CB2 variant? Is this due to the antimicrobial nature of the syrup or the addition of sugars (e.g. sucrose) that the selected yeast strain does not ferment?
• Why are the fermentation degree is missing after the second fermentation? Again, we must return to the motivation for the fermentation regime and the yeast strain chosen (see question 5). An interesting point is the reduction of the alcohol content by 0.4% in the CB2 variant at the end of the second fermentation? The yeast itself, according to the manufacturer, has almost an 80% fermentation rate.
• What is the initial extract of the three types of wort, i.e. by how much does the extract of the wort increase after the addition of the extract.
8. Please comment why there is a decrease in AOA in the third variant on the 60th day of fermentation?
Minor editing of English language required
Author Response
We thank the reviewers for their comments, their critical reading of our manuscript (MS) and for the time they have taken to provide this very helpful feedback. We have taken all the reviewer’s comments into consideration and have modified our MS accordingly. We are convinced the changes have contributed to improve our MS. To facilitate identification of the changes we have used track changes in the revised MS. 

Reviewer 2 Report
Comments and Suggestions for Authors
Fermentation of a strong dark ale hybrid beer enriched with carob (Ceratonia siliqua L.) syrup with enhanced polyphenol profile
The work carried out is interesting and it seems that it can be published in this journal, however, the authors must clarify some aspects, and if necessary, complement their studies with additional experiments.
Why didn't the authors use a control, that is, some commercial beer to compare their results, in all aspects such as quality parameters and chemical analysis? Explain or please doing experiments.
If making a beer or doing an enriched fermentation to make a beer, why didn't the authors hold a tasting panel?
Even if the parameters or chemical studies have been good, how can you know if the beer can be marketed if the flavor is not pleasant?
Title. Correct the format of the title according to the guide for authors.
Abstract. Use corrects international units for “Vol.”.
Line 29. Antioxidant activity is only an in vitro evaluation but is not a biological activity. Please correct or explain in detail.
Authors should format their graphs better and improve the quality.
Keywords: Replace repeated words in the title, instead changing them to “antioxidant activity”, “phenolic compounds”, “LC-QToF-MS analysis”, etc.
In section 2.2.2. Spectrophotometric analysis, the authors must describe the methodology in detail. It is not possible that the process was exactly as described in the cited references.
What were the concentrations used to make the calibration curve, what were your standards for the curve, what volume did you use, etc?
Comments on the Quality of English LanguageMinor editing of English language required
Author Response

(The authors gave the same response as above.)

Round 2
Reviewer 1 Report
Comments and Suggestions for Authors
The authors take into account all comments.
Author Response
We thank the reviewer for the valuable comments that has helped us improve significantly our manuscript.
Reviewer 2 Report
Comments and Suggestions for Authors
The authors improved the manuscript and considered most of the comments. However, there seem to be errors in the graphs in Figure 4. The homogeneous groups do not correspond to the size of the columns. For example, in "A) TOTAL PHENOLIC CONTENT (TPC) OF CAROB BEERS" column "B-30d" should be "a", "B-0d" should be "b", "CB1-30d" should be "c ", etc. Please verify all figures. Columns of greater or lesser size must have "a" or the last letter used. For example in "B) ANTIOXIDANT ACTIVITY OF CAROB BEERS" the column "B-60d" must have "a" and "CB2-30d" must have "h".
Please verify these data and the discussion of the results according to the homogeneous groups should also be carefully reviewed.
On the other hand, conclusion section are so long.
With these errors, the article cannot be accepted.
Comments on the Quality of English LanguageMinor editing of English language required
Author Response
The authors improved the manuscript and considered most of the comments.
Answer: We thank the reviewer for the valuable comments that has helped us improve significantly our manuscript.
However, there seem to be errors in the graphs in Figure 4. The homogeneous groups do not correspond to the size of the columns. For example, in "A) TOTAL PHENOLIC CONTENT (TPC) OF CAROB BEERS" column "B-30d" should be "a", "B-0d" should be "b", "CB1-30d" should be "c ", etc. Please verify all figures. Columns of greater or lesser size must have "a" or the last letter used. For example in "B) ANTIOXIDANT ACTIVITY OF CAROB BEERS" the column "B-60d" must have "a" and "CB2-30d" must have "h" .Please verify these data and the discussion of the results according to the homogeneous groups should also be carefully reviewed.
Answer: We thank the Reviewer for this valuable comment. We apologize for this issue and we have made all the appropriate corrections in all graphs in Figure 4 and the caption. We have verified the results.
On the other hand, conclusion section are so long.
Answer: We thank the reviewer for this comment and following this suggestion, we have deleted some phrases from the conclusion to make it shorter. Specifically, we have deleted lines 625-628 and lines 634-637.
With these errors, the article cannot be accepted.
Answer We thank the reviewer for all the valuable remarks that have contributed to the improvement of the manuscript. We believe that we have amended all errors in the manuscript.